# REAL-TIME TEXT-CONDITIONED WORLD MODELS TOWARDS INTERACTIVE PROTOTYPING

## ABSTRACT

State-of-the-art world models have been used to produce sequences of gameplay that accord with provided user-input actions, with the suggestion that such models could have creative applications such as quick prototyping of game ideas. However, high quality, consistent gameplay generation often comes at the cost of inference speed, making real-time interactive play challenging. Models are also limited in their ability to generate new content that deviates from original gameplay, particularly when trained on data from a single environment. In this work we demonstrate two major steps towards enabling interactive, real-time ideation. Building on an autoregressive world model capable of generating highly consistent and complex sequences over minutes (Kanervisto et al., 2025), we enable substantial model speed-up with minimal deterioration in output quality. This is done by replacing the next-token prediction paradigm with discrete diffusion, introducing a lightweight refinement transformer which carries out iterative masked predictions. Subsequently, we explore how new game behaviours can be learned and triggered at inference time in a controlled manner. To this end, we introduce text to control the game environment generated by the model, and curate the BodySwap dataset which simulates a character swapping mechanism allowing to change the playable character using a text prompt. Our results highlight the potential of world models as real-time prototyping tools, enabled by intentional curation of small datasets and efficient finetuning.

## 1 INTRODUCTION

In recent years, impressive progress has been achieved towards high quality video generation. Large scale models (Brooks et al., 2024; Sharma et al., 2025) trained on real-world footage, video games, and interactive 3D environments have showcased a potential to learn rich and complex environments and interactions. While text remains the main mode of interaction with video generation models, there is growing interest in action-controlled models (e.g. using navigation keys, game controller actions), often referred to as interactive video models or *world models* (Ball et al., 2025; Decart et al., 2024; He et al., 2025). Potential applications for such world models include self-driving simulation (Russell et al., 2025), robotics (NVIDIA et al., 2025), and supporting creative ideation and game development (Kanervisto et al., 2025). In the context of video gaming, models trained on large scale single game environments have demonstrated an ability to learn and faithfully reproduce game maps and physics, complex interactions and character abilities in a responsive environment (Che et al., 2024; Alonso et al., 2024). This ability to rapidly generate and interact with high fidelity new game content could prove useful for quickly prototyping new video game ideas.

Current models, however, fall short of the capabilities that would be required to fulfil this vision. Existing models suffer from two key limitations, which substantially reduce the ability to quickly test new ideas. Firstly, world models struggle to generate long videos of consistent gameplay in real time. Models capable of real time generation are often limited to simple interactions (e.g. navigation) and video quality quickly degrades after a minute of gameplay (Alonso et al., 2024). In contrast, autoregressive models like WHAM (Kanervisto et al., 2025) are capable of generating long videos with very high fidelity visuals and complex interactions, but take minutes to generate a few video frames. Secondly, single-game dedicated models are only trained on existing gameplay instances, which prevents models from generating new content that deviates from the original game. The latter could be addressed to some extent by training world models not on one game, but on

large, diverse datasets encompassing multiple games and environments (Bruce et al., 2024). This comes at the expense of control over what the model can and cannot generate, yielding a multi-purpose model with the ability to generate a large variety of interactive environments but potentially lacking in-depth understanding of complex mechanics and interactions. In addition, training on vast quantities of data, often with limited transparency as to its provenance, poses an ethical concern about the use of resources and the respect of copyright.

In this work, we adopt a different strategy to address these limitations, focusing on maximising user control over generated content. Building on the WHAM (Kanervisto et al., 2025) world model, we first adapt the model to enable real time generation by moving from a next-token prediction setup to a discrete diffusion paradigm designed to maximise efficiency. This is done by introducing a lightweight refinement transformer, which iteratively refines image predictions from a larger backbone transformer using a MaskGit-inspired (Chang et al., 2022) inference process. In order to explore prototyping with our real-time model, we propose to move away from expensive training on very large scale datasets, and demonstrate how new behaviours and modalities can be introduced in an efficient manner by finetuning on small datasets curated to simulate a new game mechanic. Concretely, we curate a dataset called BodySwap, which assembles gameplay sequences based on character location and camera angle, effectively simulating an in-game character switch. In order to control when this new mechanism comes into play, we associate character swaps with text instructions based on character name, ability or appearance. Our experiments show that our novel WHAM-RT (WHAM Real Time) model achieves more than $7000\%$ speed-up over WHAM with limited quality loss, maintaining its ability to generate coherent, long horizon videos with complex interactions. For BodySwap, we investigate different finetuning options to learn the new behaviour and introduce the text modality. Our results show that it is possible to introduce new behaviours in an efficient manner whilst preserving the world model quality.

To summarise, this work makes two key contributions with the ultimate aim of moving world models towards real time, customisable and controllable models. Our contributions are the following:

1. We introduce WHAM-RT, an efficient autoregressive world model designed to enable real-time play and interaction for users. Building on the WHAM model (Kanervisto et al., 2025), we replace the next-token prediction paradigm with an efficient MaskGit (Chang et al., 2022) inspired discrete diffusion set up. WHAM-RT achieves up to 17 frames per second and can generate consistent and accurate gameplay for at least 5 minutes.

2. We demonstrate that we can introduce controlled new behaviours and mechanisms in a pre-trained world model through efficient model adaptation. This approach is centred around intentional curation of small datasets and the introduction of the text modality, allowing the user to quickly iterate on the design of new behaviours and control when they are triggered.

With this work, we aim to highlight the possibility of small and intentional dataset curation for creative prototyping. We hope to motivate future work to explore controllable approaches to generalise to new environments using methods that can leverage limited amounts of data. To further foster research, we plan to make our WHAM-RT model and BodySwap dataset publicly available.

## 2 RELATED WORK

**Video Generation.** Video generation has evolved along two dominant paradigms, each with distinct architectures and trade-offs. Diffusion-based approaches, particularly Latent Diffusion Models (Rombach et al., 2022) with Diffusion Transformers (DiT) (Peebles & Xie, 2023), achieve strong perceptual quality and temporal stability (Ho et al., 2022; Blattmann et al., 2023; Esser et al., 2024; Wan et al., 2025). However, their iterative denoising process requires multiple function evaluations, fundamentally limiting inference speed despite recent advances in distillation and consistency models (Salimans & Ho, 2022; Song et al., 2023; Sauer et al., 2024b;a). Autoregressive approaches, inspired by large language models (Minaee et al., 2024), discretize video into token sequences using VQ-VAE or VQ-GAN (Van Den Oord et al., 2017; Esser et al., 2021) and generate them with causal transformers. Models like VideoGPT (Yan et al., 2021), VideoPoet (Kondratyuk et al., 2024) and WHAM (Kanervisto et al., 2025) demonstrate precise sequence control but face two critical limitations: sequential decoding bottlenecks that prevent real-time inference, and error accumulation over long horizons. Recent works on non-autoregressive masked prediction, such as MaskGit, MAGVIT

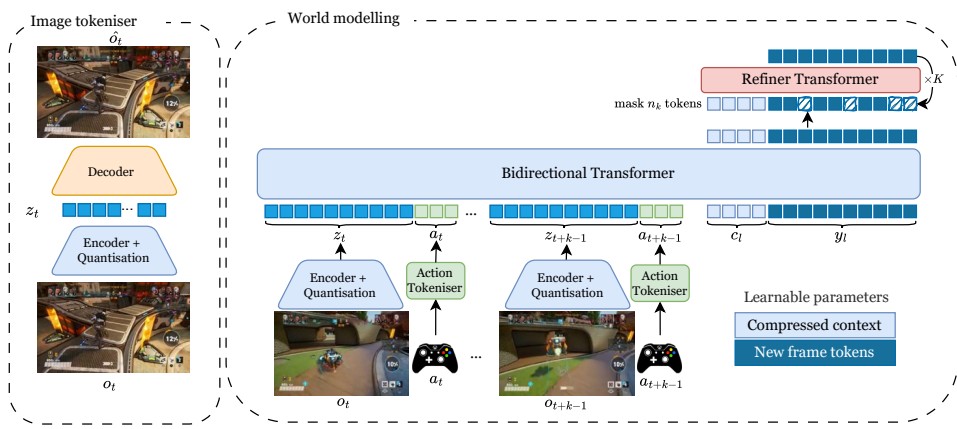

Figure 1: Overview of the WHAM-RT architecture.

and VAR (Chang et al., 2022; Yu et al., 2023; Tian et al., 2024), explore parallel token generation to address these bottlenecks. We take inspiration from these approaches to increase the generation efficiency of autoregressive discrete world models (Kanervisto et al., 2025).

**Controllable Generation.** Current controllable generation methods inject control signals at different granularities. Global guidance through text prompts uses cross-attention or adaptive normalisation, while spatial-structural signals (pose, depth) typically require adapter modules (Zhang et al., 2023). Specialised methods like CameraCtrl and MotionCtrl (He et al., 2024; Wang et al., 2024) provide trajectory-level control, and world models condition on actions to capture agent-environment dynamics (Bruce et al., 2024; Kanervisto et al., 2025; Russell et al., 2025). The primary limitation across these approaches is their requirement for extensive per-modality training on large scale datasets. Each new control signal demands substantial training, creating significant friction for rapid prototyping scenarios where novel behaviours must be quickly tested. Our efficient approach demonstrates that new control modalities, such as text-triggered character swapping, can be efficiently learned from minimal curated data and deployed without full model retraining, enabling iterative creative exploration.

**World Modelling.** Generative world models go beyond conventional video synthesis by requiring accurate, per frame responsiveness to control signals while maintaining physical plausibility and long-range consistency. Domain-specific progress includes autonomous driving simulators (GAIA-2) (Russell et al., 2025), interactive gameplay generators (the Genie family) (Bruce et al., 2024), and controllable game environments (WHAM) (Kanervisto et al., 2025). Other efforts like WorldMem (Xiao et al., 2025) add explicit memory to diffusion backbones to sustain coherence and GameN-Gen couples reinforcement learning with diffusion for gameplay generation (Valevski et al., 2024). Real-time performance remains challenging and current methods require complex post-training distillation steps (Guo et al., 2025; Lin et al., 2025; Chen et al., 2024) to reduce the number of denoising steps or introduce sparse attention patterns (Zhang et al., 2025), often at the expense of complex interactions and long horizon coherence. Models trained on limited and single game environments like MineWorld Lin et al. (2025) struggle to generate novel content beyond their training distribution, limiting creative applications. Our work addresses those two challenges by enabling real-time world modelling, while our BodySwap experiment demonstrates that world models can efficiently learn and express novel behaviours not present in the original training data.

## 3 PRELIMINARIES: WHAM

WHAM (Kanervisto et al., 2025) is an autoregressive causal transformer model trained on a large dataset of human gameplay data from the 3D multiplayer video game Bleeding Edge. The data was collected in partnership with the Ninja Theory game studio, with the players' consent, and it amounts to roughly 500,000 anonymized gaming sessions. Given a discretised series of interleaved tokens for video frames and corresponding controller actions, the model learns to predict the tokens

of the next video frame and action in the sequence. The model uses a VQ-GAN Esser et al. (2021) encoder/decoder trained from scratch on the same Bleeding Edge dataset and uses learned position embeddings. Controller actions are encoded using a simple action tokenizer into two sets of tokens: a set of multi-hot encodings of button presses, and discretised $(x, y)$ coordinates of the controller joysticks that control the player and camera direction.

Bleeding Edge comprises complex multiplayer and object interactions, as well as unique character abilities and special attacks. WHAM has demonstrated an ability to produce impressively consistent long sequences of gameplay, including character-specific abilities and interactions with non-playable characters. However, generating each frame is a slow process, making it unwieldy to use in practice in an interactive, playable setting, where the user will expect an instant reaction to their inputs. In this work, we introduce WHAM-RT (WHAM Real Time), which uses an alternative architecture in order to offer a similar experience to WHAM, but with much faster inference, thus enabling a truly responsive experience. This improved generation speed is a crucial development if these models are to be used in practical applications such as prototyping in the future.

## 4    WHAM REAL-TIME

In order to accelerate the inference process of WHAM-like autoregressive models, we propose replacing the next-token prediction paradigm (Kanervisto et al., 2025) with an iterative masked-and-prediction inference process, inspired by the MaskGit (Chang et al., 2022) formulation for image generation. In contrast to predicting one image token at a time, MaskGit aims to iteratively predict a subset of masked image tokens, starting from a fully masked image and iteratively reducing mask coverage until the full set of visual tokens is recovered. This strategy substantially accelerates inference, and was successfully leveraged in large foundation models for fast image (Chang et al., 2023) and video generation (Villegas et al., 2023). Despite such speed gains, the foundation MaskGit models still require multiple iterations of prediction through a large transformer model, preventing them from reaching real-time performance.

**Model architecture.** We design WHAM-RT to address these limitations, aiming to strike a balance between leveraging the predictive power of large autoregressive transformers and the benefits of iterative masked refinements, while at the same time maximising inference speed. Illustrated in Figure 1, WHAM-RT takes as input context a sequence $S$ of $N$ interleaved image and action tokens, tokenised following Kanervisto et al. (2025). Concretely, WHAM-RT operates in the discrete latent space of a pre-trained VQ-GAN model (Esser et al., 2021). The model architecture comprises a large backbone transformer $\mathcal{B}_t$ that generates a first estimate of the tokens of the next image in the sequence. Similarly to MaskGit, our backbone transformer has bidirectional attention, and predicts all new image tokens *in parallel*. To achieve real-time inference, we introduce a lightweight refinement transformer $\mathcal{R}_t$, which iteratively updates predicted image tokens using MaskGit iterations.

**Video Context Compression.** In order to handle the two-level architecture, and the lack of iterative masking in the backbone transformer, we introduce a set of learnable input tokens $T_\ell = \{\mathbf{c}_\ell, \mathbf{y}_\ell\}$, where $\mathbf{y}_\ell$ is the set of input image tokens, such that $\mathbf{y_b} = \mathcal{B}_t(\mathbf{y}_\ell | \mathbf{S}, \mathbf{c}_\ell)$ are predicted image tokens of the next frame in sequence $S$. Tokens $\mathbf{c}_\ell$ play a crucial role towards achieving real time performance of learning a compressed representation of context $S$. Similar to the concept of Registers (REG) in vision transformers (Darcet et al., 2024), $\mathbf{c}_\ell$ are a new set of learnable tokens introduced to summarise and compress the provided context, allowing fast refinement iterations. Concretely, we compute the compressed context as $\mathbf{c_b} = \mathcal{B}_t(\mathbf{c}_\ell | \mathbf{S}, \mathbf{y}_\ell)$. The masked iterative refinement is then carried out as $\mathbf{y_r} = \mathcal{R}_t(\mathbf{y_b} | \mathbf{m}, \mathbf{c_b})$, where $m$ is the mask that describes the tokens that are updated.

**Training.** We train the model by computing the cross-entropy loss between predictions from both backbone and the refinement transformer and the ground truth image tokens $\mathbf{y}$. For the refinement transformer, we adopt a teacher forcing approach where ground truth tokens are provided as input. We randomly mask a percentage $p$ of tokens such that $0 < p \leq p_{max}$ and compute the loss of masked tokens only. As Kanervisto et al. (2025) demonstrated the benefits to predicting images and actions together, and to ensure training procedures are as similar as possible, we further introduce an action prediction auxiliary loss using a separate lightweight transformer head $\hat{\mathbf{a}} = \mathcal{A}_t(\mathbf{y}, \mathbf{c_b})$. Defining $\mathbf{a}$ as the ground truth action, our overall loss function is therefore:

$$\mathcal{L} = \mathcal{L}_{CE}(\mathbf{y}, \mathbf{y_b}) + \mathcal{L}_{\mathbf{CE}}(\mathbf{y}, \mathbf{y_r}^{\mathbf{y_b}, \mathbf{P}}) + \mathcal{L}_{\mathbf{CE}}(\mathbf{a}, \hat{\mathbf{a}}) \qquad (1)$$

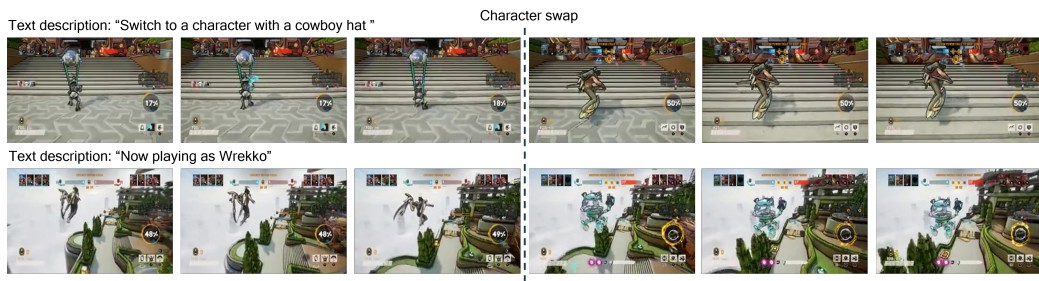

Text description: "Switch to a character with a cowboy hat"

Character swap

Text description: "Now playing as Wrekko"

Figure 2: Visual examples of the BodySwap dataset

**Inference.** The generation process is carried out in a frame-level autoregressive manner. Given a tokenised context of $N$ interleaved images and actions, we predict the next image by computing a first estimation of image $\mathbf{y_b}$ and compressed context tokens $\mathbf{c_b}$ using the backbone transformer. The refinement transformer then iteratively updates the image tokens for $M$ iterations. Starting at $p_0 = p_{max}$, we iteratively reduce the amount of masked tokens following a simple linear decay function: $p_m = p_{max} \cdot (1 - \frac{m}{M})$. For both transformers, we apply the softmax function to predictions with separate temperature values $T_b, T_r$. Once a frame is predicted, we shift the context by integrating the new frame and predict the next frame in the sequence.

## 5 LEARNING NEW TEXT CONTROLLABLE MECHANICS

Achieving real-time inference is a significant step forward towards using world models to test the gameplay experience and responsiveness of new ideas. In order to ultimately create playable proto-types, it is additionally necessary to introduce new game behaviours in a controllable manner. We propose to achieve this using small training datasets intentionally curated to simulate new game mechanics, and introducing text to trigger these learned new behaviours on the fly. Concretely, we focus on using text to exert control over the character being controlled by the player, creating a model that can swap between characters based on a diverse range of prompts. This approach could subsequently be extended to other elements of gameplay, such as items present in the environment, the map currently being played, or even new characters and items.

**BodySwap Dataset.** We build BodySwap on top of the WHAM and WHAM-RT's training dataset (see Appendix F.1), comprising recorded gameplay sequences of the Bleeding Edge game. The game features 13 different characters, each with their own distinct appearance and abilities. Swapping between characters in game is not an existing game mechanism and had to be simulated. This was achieved by stitching together pairs of gameplay segments featuring two different characters, such that one "bodyswap" example comprises $K$ frames as character A, followed by $K$ frames as character B. Visual examples of our dataset are shown in Figure 2. The main challenge building the dataset was finding sequence matches that were close enough to simulate character swapping behaviours. We sought matches in *symbolic* space rather than *visual* space, and matching candidates were further filtered to ensure a uniform coverage of both the playable map and possible character swaps. More details on the dataset construction are available in Appendix F.2.

Lastly, we built swapping text instructions based on 1) character names (e.g. "Swap to Azrael"), 2) character visual attributes (e.g. "Swap to a character who has wings"), and 3) character in game ability (e.g. "Swap to a character who can fly"). After collecting character attributes, prompt instructions were generated on the fly during training using templates constructed via a handcrafted context-free grammar. Details on prompts construction are available in Appendix D.

**Introducing The Text Modality.** This task requires introducing a new modality to a model during the process of finetuning. Since the model has not previously been trained on text, it has no initial basis from which to infer the meaning of the prompts. In order to give the model a starting point in interpreting the text, we use a frozen pre-trained language model to encode the prompts before passing them to the model as input. We additionally introduce a learnable linear layer to convert text embeddings to the model's latent space. Our modified architecture is shown in Figure 3. We introduce text inputs in sequence, such that context has the following structure: $S =$

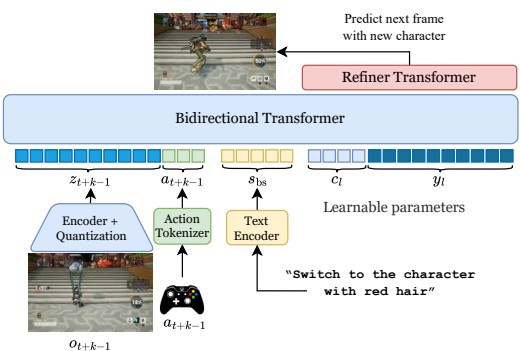

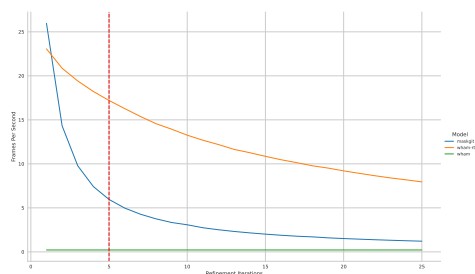

Figure 3: Overview of the BodySwap enabled WHAM-RT model.

Figure 4: Model FPS calculated using a batch size of 1 on a single H100 accelerator.

[IMG (CHR A)][ACT][IMG (CHR A)][ACT][TEXT][IMG (CHR B)][ACT][IMG (CHR B)]. No changes are made to the refinement transformer.

**Finetuning Procedure.** Successfully training a BodySwap model requires learning the new character swap mechanic (successfully swapping *and* preserving the character) while at the same time maintaining the world model capability of WHAM-RT. Finetuning the model to only learn to predict the swapped character frame after a text input could lead to catastrophic forgetting, where a model constantly swaps across characters. We adopt a robust procedure, randomly sampling the starting frame from $N$ frames before the swap (no swap or text in the $N$ frame sequence), to $N$ frames after the swap ($N$ frames following a text instruction). This allows us to introduce enough diversity in swap location and learn to maintain a character swap, while at the same time preserving the world model by seeing swap free training samples. We additionally introduce a text curriculum, where training starts with simplest prompt structures (e.g. `"Swap to CHAR"`) which get increasingly complex as training progresses. This is done in particular to allow learning of character names first, then introduce character attributes.

**Efficient Adaptation.** In addition to full finetuning, we consider parameter efficient finetuning (PEFT) methods (Hu et al., 2022; Liu et al., 2024b), which enable substantially reducing training costs as well as minimising the impact of learning the new behaviour on world model ability. At inference time, PEFT layers are only activated when a text condition is inputted into the model, allowing to fully preserve world modelling ability when not swapping characters.

## 6 EXPERIMENTS

**Implementation details.** We keep WHAM-RT's architectural design as close as possible to WHAM (Kanervisto et al., 2025) so as to focus our evaluation on the impact of changes made for efficient inference. We use the pre-trained VQGAN model from Kanervisto et al. (2025) and keep it frozen. Both backbone and refinement transformers use a GPT2 architecture (Radford et al., 2019) with 24 attention heads and embeddings of dimension 1536. The former comprises 16 layers for a total of 450M parameters, while the latter has 8 layers and 225M parameters. The auxiliary head for action prediction uses a GPT2 causal transformer with 4 layers. The compressed context tokens comprise 512 tokens. We set $p_{max} = 0.55$ for refinement iterations. The model has a context of 9 interleaved images (540 tokens) and actions (16 tokens). We train the model for $200k$ steps with mixed precision using FSDP (Zhao et al., 2023) and batch size 600. We use the same Bleeding Edge dataset as in (Kanervisto et al., 2025) to train WHAM-RT. BodySwap models are trained for $100k$ steps and use BERT (Bertasius et al., 2021) as our frozen text encoder. For PEFT models, we consider LoRA (Hu et al., 2022) and variant DoRA (Liu et al., 2024b), with rank and alpha set to 64. PEFT layers are introduced in self attention layers and the first projection layer of the MLP blocks. The BodySwap dataset comprises 41k training examples. Both Bleeding Edge and BodySwap training datasets are downsampled to 10 Hz. For all experiments, we use $\mathcal{T}_b = 0.7$, $\mathcal{T}_r = 0.5$ and $M = 5$ unless specified otherwise. To measure the quality of generated gameplay, we use a set of 1024 Bleeding Edge

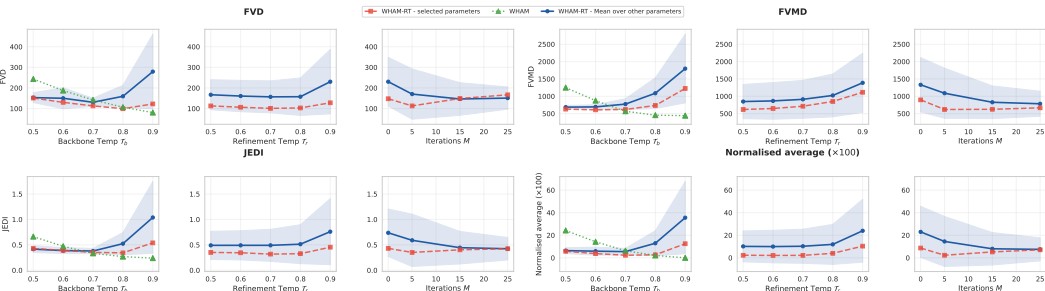

Figure 5: Impact of temperatures and iterations on FVD, FVMD, Jedi and normalised average values. For a given parameter and metric, we report mean and standard deviation values with respect to other parameters. We additionally show values for our default parameters (red line), and WHAM (green line). For all metrics, lower is better.

examples as ground truth. We use the first 9 frames and action sequences from these samples as input to the model to generate 1024 test gameplay videos comprising 100 frames.

**Human Evaluation.** We conduct a controlled, human evaluation to compare our models on two different tasks: world modelling performance and BodySwap quality. For each task, annotators viewed side-by-side A/B videos of the same clip with blinded model type and randomised left/right order. For the general world modelling task we evaluate the overall best-looking video and the correct executions of the actions, guided by a step-by-step action display. For the BodySwap task, we evaluate the quality of the swap, the identity correctness and the temporal persistency. We report human A/B judgments from 9 different participants aggregated across all clips and raters for two tasks. Full questions, UI details and additional detailed results are provided in Appendix E.

## 6.1 WHAM-RT RESULTS

**Quantitative evaluation.** To evaluate WHAM-RT's performance, we focus our evaluation on 1) the quality of generated gameplay and 2) generation speed by measuring Frames Per Second (FPS). For quality, we compute three different metrics, measuring different aspects of generation performance. Firstly, similar to Kanervisto et al. (2025), we compute the Fréchet Video Distance (FVD) (Unterthiner et al., 2019), together with a more robust variant, Jedi (Luo et al., 2024), as a measure of statistical similarity between distributions of generated videos and ground truth. We additionally measure motion accuracy by computing the Fréchet Video Motion Distance (FVMD) (Liu et al., 2024a), which compares motion vectors between ground truth and generated results. The latter was reported to be closer to human judgement than FVD-type metrics in Russell et al. (2025).

In Figure 5, we can see all metrics, compared to WHAM, for different values of backbone temperature $\mathcal{T}_b$. We can see that while WHAM consistently improves when $\mathcal{T}_b$ increases, WHAM-RT favours lower temperatures, with more instability at high temperatures. Results show that WHAM-RT's performance, in terms of consistency (FVD, Jedi) shows a small performance loss compared to WHAM's best configuration. More differences are observed for FVMD, suggesting reduced motion consistency. This could be attributed to the compressed context used to refine visuals, potentially introducing jitters. Nonetheless, differences with respect to WHAM remain small, highlighting that WHAM-RT's significant speedup is achieved at limited performance loss.

To measure generation speed, we compute FPS for different numbers of refinement iterations $M$, as these directly impact speed. We compare to next token prediction (WHAM), as well as MaskGit inference (i.e. carrying out refinement iterations within the backbone transformer). We report results in Figure 4. We can see that WHAM-RT achieves substantial generation speedup reaching up to 17 FPS with 5 refinement iterations. This is an acceleration of up to 7895% compared to WHAM, allowing users to interact with the model in real time. Compared to a MaskGit model, the acceleration is also significant. This comparison notably highlights that the smaller architecture of WHAM-RT is not sufficient for real-time inference, supporting the need for our lightweight refinement transformer.

**Ablations.** In addition to the backbone temperature, we investigate the influence of two key parameters: the refinement temperature $\mathcal{T}_r$ and iterations $M$. We generate gameplay sequences for all 1024

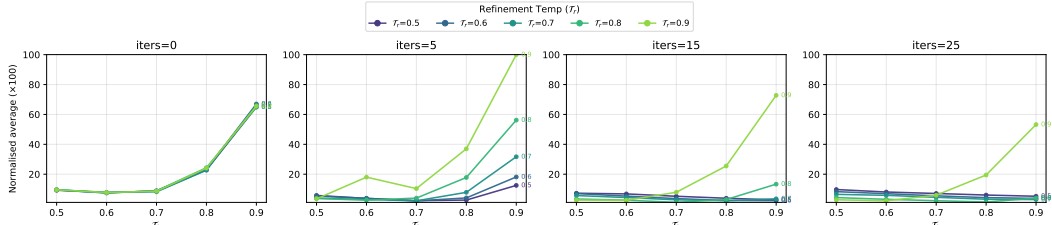

Figure 6: Impact of temperature parameters on the normalised average of computed metrics (FVD, FVMD, Jedi) per number of iterations. Per metric results are available in Appendix B.

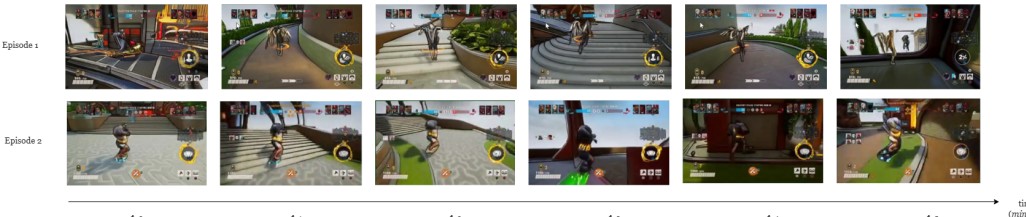

Figure 7: Generated frames taken each minute from 2 different long gameplays with WHAM-RT.

test examples using 100 different configurations, with $\mathcal{T}_r$ and $\mathcal{T}_b \in [0.5-0.9]$ and $M \in [0, 5, 15, 25]$. Results in Figures 5 and 6 show that the backbone temperature has the largest influence over performance, with better quality results at low temperatures and largest temperatures substantially altering generation quality. The refinement temperature shows a similar, but more stable behaviour. Lastly, larger numbers of iterations yield results that are more robust to temperature values, however lower (non-zero) number of iterations achieve the best overall performance at low temperatures.

**Long horizon coherence.** In this experiment, we test WHAM-RT's ability to generate frames across an extended time horizon. We ask 2 human participants to generate 5-minute long recordings of interacting with the WHAM-RT model by inputting games actions via either a game controller or using a keyboard. In Figure 7, we demonstrate the model's ability to coherently preserve the game mechanics, visuals and physics across an extended time horizon, by showing video frames across different time intervals. The full 5 minute long videos are provided in the supplementary material.

**Human evaluation results** World modelling ability is measured as the fraction of pairwise trials in which a model was preferred by human participants. Percentages are computed per model as wins divided by the total number of comparisons. Results are shown in Figure 9a and 9b. The WHAM and WHAM-RT models rank significantly higher in terms of human preference, compared to the fully fine-tuned BodySwap model. As expected, WHAM is often the preferred model against WHAM-RT. Nonetheless, WHAM-RT still achieves robust performance, with human raters also highlighting the controllability and accuracy of the executed action.

## 6.2 BODYSWAP RESULTS

**Quantitative evaluation.** To evaluate the accuracy of our new BodySwapping mechanism, we follow the protocol of Hendriksen et al. (2025) and fine-tune the projection head of a PaliGemma vision-language model (VLM) (Beyer et al., 2024) to identify the character on screen by answering the question "Does the image show character A?" with a yes/no answer. As shown in Hendriksen et al. (2025), this approach can achieve over 99% prediction accuracy, providing a robust solution to evaluate our BodySwap models. We build a test set of 165 ground truth examples, with a uniform distribution of characters swapped to and from, and generate gameplay sequences, with a character swap, for the fully fine-tuned (Full), LoRA and DoRA trained models. To ensure accurate results, all prompts correspond to a single character.

Results are reported in Figure 8. We use our fine-tuned VLM on each generated frame, and report 1) Swap success rate (number of generated frames where the right character is recognised), 2) Swap

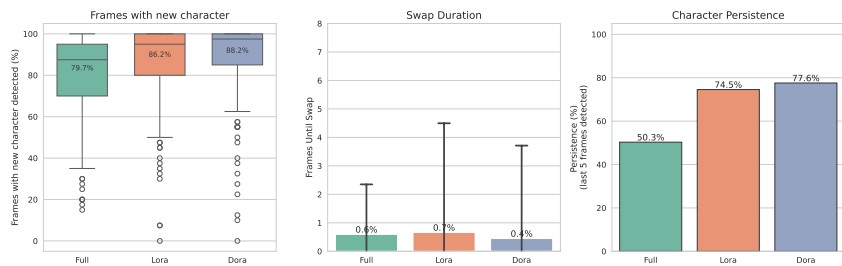

Figure 8: Quantitative BodySwap results. We measure *Swap Success Rate*: number of generated frames with the target character, *Swap Duration*: number of frames until the target character is recognised, and *Character Persistence*: target character present in the last 5 frames.

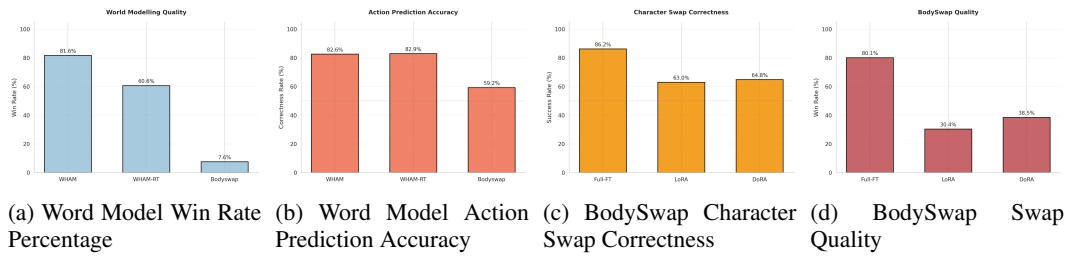

(a) Word Model Win Rate Percentage
(b) Word Model Action Prediction Accuracy
(c) BodySwap Character Swap Correctness
(d) BodySwap Swap Quality

Figure 9: Human Evaluation Results

duration (number of frames needed to first identify the new character), and 3) Swap persistence (is the new character still recognised at the end of the video). We can see that all models achieve a robust performance, with a high overall swap success rate, and nearly immediate swaps on average. Our results show PEFT models achieving the best performance, with overall best results obtained by the DoRA model. This can be partially attributed to the degradation in world model quality caused by full fine-tuning, making it more challenging for the VLM to recognise characters.

**Human evaluation results.** The human evaluation study focused on measuring the character-prompt accuracy and quality of the swap between the 3 fine-tuning choices: full fine-tuning (fullFT), LoRA and DoRA. Figure 9d shows the win-rate of the different models, while Figure 9c shows the recorded swap success rate. Interestingly, results contrast with our quantitative results, showing that the full fine-tuning model outperforms DoRA and LoRA models, with the DoRA model outperforming LoRA. This suggests that the fullFT model achieves more accurate swaps, at the cost of a poorer world modelling ability. In contrast, LoRA and DoRA models have the potential to fully maintain world model ability, by turning off new layers after the swap.

## 7 CONCLUSIONS AND FUTURE WORK

In this work, we introduce two key solutions towards leveraging world models for interactive prototyping and creative ideation. Firstly, we introduce WHAM-RT, an adaptation of highly consistent world model WHAM (Kanervisto et al., 2025), enabling real-time inference through a two-stage discrete diffusion modelling approach. Secondly, we demonstrate how new capabilities and modalities can be introduced to a pre-trained model cheaply, both in terms of data requirement and training costs. Our work seeks to emphasise the potential for intentional, high quality and efficient data curation and multi-modal conditioning to introduce new ideas and behaviours in world models in an intuitive and controllable manner. Achieving substantial acceleration did incur a reduced world modelling quality compared to WHAM, future work will explore improved model architectures, as well as alternative, more efficient, image generation approaches (Tian et al., 2024; Li et al., 2024) to further improve generation quality and speed. Building on learnings from the BodySwap task, we will explore more complex tasks, environment modifications, and conditioning forms; while at the same time continuing to improve model adaptation techniques.

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

# A    VIDEO RESULTS

As supplementary material, we include representative generated video examples from following models used in our experiments:

- WHAM: 3 examples, length: 11 seconds each
- WHAM-RT: 3 examples, length: 11 seconds each
- WHAM-RT (long time horizon videos): 2 examples, length: 5 minutes each
- BodySwap Full Finetuning: 15 examples, including 5 failure modes, length: 5 seconds each
- BodySwap LoRA: 6 examples, length: 5 seconds each
- BodySwap DoRA: 6 examples, length: 5 seconds each

# B    ADDITIONAL QUANTITATIVE RESULTS

## B.1    WHAM-RT METRICS

In this section, we provide detailed results for FVD, FVMD and Jedi metrics, as well as their normalised average, computed by averaging normalised metrics. We provide iteration-stratified heatmaps in Figure 10, showing metrics computed for all configurations. Jedi and Normalised average values were multiplied by 100 to improve readability. In addition to Figure 6, which reports results for the normalised average, we show the impact of refinement temperature and number of iterations over all individual metrics in Figure 11, with respect to the backbone temperature.

We can see that all metrics become more stable at higher temperature with more iterations, with FVD and Jedi preferring high refinement temperatures, while FVMD achieves consistently better results at low temperatures. This can be attributed to the fact that the metrics measure different properties of the video, with FVD/Jedi measuring visual similarity, while FVMD measures motion accuracy. Visually, we have observed more lively environments (e.g. more NPCs) at high temperatures, at the expense of more flickering and instability, which supports these observations.

## B.2    BODYSWAP ADDITIONAL RESULTS

In this section, we report additional BodySwap results, in the situation where the input prompt is ambiguous, e.g. when multiple characters are valid targets (e.g. "Swap to a female character"). We collect a small set of 21 ambiguous test examples and generate gameplay sequences. To evaluate the accuracy of the swap, we collect, for each prompt, the set of valid target characters. We then query the VLM with a prompt per valid character, and keep as target the first character with a non-zero prediction score. If no character is detected, we set results to 0 for frame detection success and character persistence, and 40 (the maximum number of frames) for swap duration.

Results in Figure 12, show that the fully fine-tuned model achieves much more robust results, with the LoRA model in particular achieving the largest performance drop compared to non ambiguous

prompts. We note that ambiguous prompts were part of our human evaluation, which can further explain the dominance of the Fully fine-tuned model over LoRA and DoRA models. We highlight the small sample size for this experiment, compared to our main paper results.

## C STATEMENTS

**Large Language Model usage.** LLMs were used to assist with the generation of plotting code, and to prototype some small aspects of the training code. All generated code was carefully checked by a human afterwards. LLMs were not used for ideation or for writing assistance.

**Ethics Statement.** Our experiments are conducted on a dataset that has been collected with the consent of the game studio and the participating users, where any personally identifiable or sensitive data has been anonymized. In all human evaluations on the new models introduced and their benchmarks, we have had the consent of the people submitting the responses.

We are aware that world modelling technology, particularly in the context of gaming, could have an impact on certain types of creative employment. We carry out our research and model design with end users in mind, seeking to provide tools that seek to enhance, rather than replace, their creative process.

**Reproducibility Statement.** Sharing work that is reproducible and can inspire the community is of key importance to us and makes our research much more valuable. In order to foster research on world models, we intend to make our work as reproducible as possible, by making our WHAM-RT model and inference code publicly available, as well as our BodySwap training data and test set.

## D DIVERSE PROMPT GENERATION

In order to generate a diverse set of possible prompts, we used a procedure involving a handcrafted context-free grammar. A context-free grammar is a tuple $G = (N, \Sigma, \mathcal{R}, S)$, where $N$ is a set of non-terminals, $\Sigma$ is a set of terminals which make up the content of generated sentences, $S \in N$ is a distinguished start symbol and $\mathcal{R}$ is a set of rules of the form $N \to (N \cup \Sigma)^*$, i.e. rules describing how non-terminal symbols can be replaced with some combination of non-terminal and terminal symbols. Sampling a sentence from a context-free grammar proceeds by beginning with the start symbol $S$ and subsequently using rules from $\mathcal{R}$ to replace symbols until no non-terminals remain. A generation of this nature can easily be represented using a tree structure.

We wished to generate a set of templates for prompts in which characters could be described by their names, and/or by their physical appearance, abilities or other characteristics. By using a relatively simple handwritten context-free grammar, we are able to capture a number of ways to describe a specific character, with a diversity of phrasing, combinations of attributes being used and different orders of sentence components. Each production rule in the grammar is marked to indicate a complexity level, and generated templates are categorised by the maximum complexity level present in their tree, to allow prompts to be structured in a curriculum. A total of 1,897 prompt templates are sampled in this manner. These templates are then filled in using known information about the characters including alternative names, abilities, attributes and appearance.

Examples of trees showing two similar generations from the grammar are shown in fig. 13. These produce the templates "This time I want to play as \$CHAR\$ who can \$ABILITY\$" and "I'd like to be \$CHAR\$ who is the one with \$VISUAL\$ now".

# E   DETAILED HUMAN EVALUATION PROTOCOL AND RESULTS

In this section we present additional details regarding our human evaluation. We collected data from $N = 10$ raters who provided informed consent. Below we report the list of questions for each task:

**Task 1: Evaluating BodySwap (10 pairs)**    For each pair, answer the following:

1. **Q1 (Smoothness).** Which video has the smoother body swap?
   *Response options:* A / B

2. **Q2 (Identity correctness, Video A).** Did *Video A* swap to the intended target character?
   *Response options:* Yes / No

3. **Q3 (Identity correctness, Video B).** Did *Video B* swap to the intended target character?
   *Response options:* Yes / No

4. **Q4 (Persistence, Video A).** Did the swap in *Video A* persist until the end of the clip?
   *Response options:* Yes / No

5. **Q5 (Persistence, Video B).** Did the swap in *Video B* persist until the end of the clip?
   *Response options:* Yes / No

6. **Q6 (Background preservation).** Which video better preserved the background (camera, lighting, other objects) after the swap?
   *Response options:* A / B

**Task 2: Evaluating World Modelling (10 pairs)**    For each pair, answer the following:

1. **Q1 (Overall quality; primary).** Which video looks best overall?
   *Response options:* A / B

2. **Q2 (Action correctness).** Are the actions executed correctly? (Use the per-timestep action display as a guide.)
   *Response options:* Both correct / Only A correct / Only B correct / Neither

The study was implemented as a custom web application in Gradio[1]. A/B randomisation and the corresponding mapping to true model identities were logged per trial to enable unbiased aggregation. An example of the user interface is shown in Figure 16.

We provide detailed results relative to different questions of the human evaluation. For the BodySwap evaluation, we include plots for (ii) background preservation wins (Figure 14a), and (ii) per-model rates of temporal persistence (14b), each computed as the proportion of trials in which the criterion was satisfied for that model. We also visualize pairwise win-rate heatmaps (Figure 15) for both world modelling and BodySwap (row vs. column = win rate of row over column), where each cell is the percentage of wins over the total head-to-head trials for that pair of models. Detailed results confirm trends discussed in the main paper.

# F   ADDITIONAL DATASET DETAILS

## F.1   BLEEDING EDGE DATASET

Data for WHAM-RT was provided via a partnership with *Ninja Theory*, who collected a large corpus of human gameplay data for their game *Bleeding Edge* - a 4v4 battle arena game. Image data was rendered in MP4 format at 60fps with a resolution of 300 x 180, alongside binary files containing the associated controller actions. A timecode extracted from the game was stored for each frame, to ensure actions and frames remained in sync at training time.

WHAM-RT was trained on the `Skygarden` game map, a set of 66,709 individual player trajectories with an average length of around nine minutes each. After applying an 80:10:10 split for training / validation / testing, and downsampling to 10Hz, this gave us approximately 310M frames (about one year of game play).

---

[1] https://www.gradio.app/

## F.2    DETAILED BODYSWAP DATA CONSTRUCTION

We provide more details on the construction of the BodySwap dataset in this section. BodySwap was constructed using the `Skygarden` dataset as a starting point. Given a set of trajectories $\mathcal{T}$ of gameplay footage featuring character A, and a set of trajectories of gameplay footage featuring character B, look for candidate frames $F_A$ and $F_B$ that match each other as closely as possible. Assuming such a match has been found, where $F_A = \mathcal{T}_A[\text{timestep}_A]$ and $F_B = \mathcal{T}_B[\text{timestep}_B]$, we can then extract two pairs of before and after sequences of length K:

$$A_{\text{pre\_swap}} = \mathcal{T}_A[\text{timestep}_A - K : \text{timestep}_A]$$
$$A_{\text{post\_swap}} = \mathcal{T}_A[\text{timestep}_A : \text{timestep}_A + K]$$
$$B_{\text{pre\_swap}} = \mathcal{T}_B[\text{timestep}_B - K : \text{timestep}_B]$$
$$B_{\text{post\_swap}} = \mathcal{T}_B[\text{timestep}_B : \text{timestep}_B + K]$$

Two samples can then be created by stitching together the opposing pairs of subsequences and labelling them:

"Character A to Character B" $= A_{\text{pre\_swap}} + B_{\text{post\_swap}}$

"Character B to Character A" $= B_{\text{pre\_swap}} + A_{\text{post\_swap}}$

A pair of frames $F_A$ and $F_B$ were considered to match each other if the player's position, health and direction of travel, and the camera's position, closely matched at those points. To avoid biases in the resulting data, the game map was divided into cells, and matches were sought for each possible pairing of characters, for each map cell. This gave us a dataset of 41260 training samples, 32576 validation samples, and 32732 test samples. Data partitions were aligned with WHAM-RT's training set. The overall balance of character swaps in the training dataset is shown in Figure 17.

We chose a value of K=10, giving us ten frames before the swap and ten frames after, or an overall sequence length of two seconds (at ten frames per second).

To ensure the dataset contained a uniform coverage of both the map and the possible character swaps, we adopted the following approach to finding the matching candidate frames $F_A$ and $F_B$:

1. Split the game map into cubic chunks, by quantizing according to the x/y/z location.

2. Using the symbolic data available, for each chunk, build up a list of every sample in the dataset which took place in that chunk, recording the character name, position, camera position, character health, actions, and the trajectory / timestep of that sample.

3. Turn each sample into a 9-dimensional point consisting of the player's x, y, z position, the camera's x, y, z position, the player's health, their y direction (forwards/still/backwards), and their x direction (left/still/right).

4. For each *character* in each chunk, build a KD-Tree from these points yielding roughly num_character x num_chunks separate trees. We note that there were parts of the map which could only be visited by a subset of characters possessing certain abilities.

5. For each chunk in turn, compile a list of the characters that visited the chunk, and, for each possible pairing of characters (character A, character B, where A != B), using the KD-Trees, find the closest pair of points $P_A$ and $P_B$ such that the Euclidian distance between them is minimised.

6. Look up the source trajectories and timesteps that yielded points $P_A$ and $P_B$, and then proceed to extract the subsequences and stitch them together, as detailed above.

With this, we were able to generate a collection of simulated swaps which, for each valid location on the map, contained an example of every valid character swap. Swaps that were excessively poor (where the Euclidian distance between $P_A$ and $P_B$ was over a certain threshold) were filtered out, but this did not significantly affect the overall balance of character swaps in the dataset.

## G BODYSWAP QUALITATIVE RESULTS

In addition to videos examples, we provide comparative visualisation of BodySwap results between LoRA, DoRA and Full Finetuning (Full FT) models. To facilitate assessment of swap success, we have selected examples with prompts containing easily identifiable character attributes (e.g. riding a motorcycle). Results are provided in Figures 18, 19, 20 and 21, confirming our quantitative observation that LoRA and DoRA models struggle more with ambiguous prompts ("ginger hair" can correspond to multiple characters) and that fully fine-tuned transitions are generally smoother.

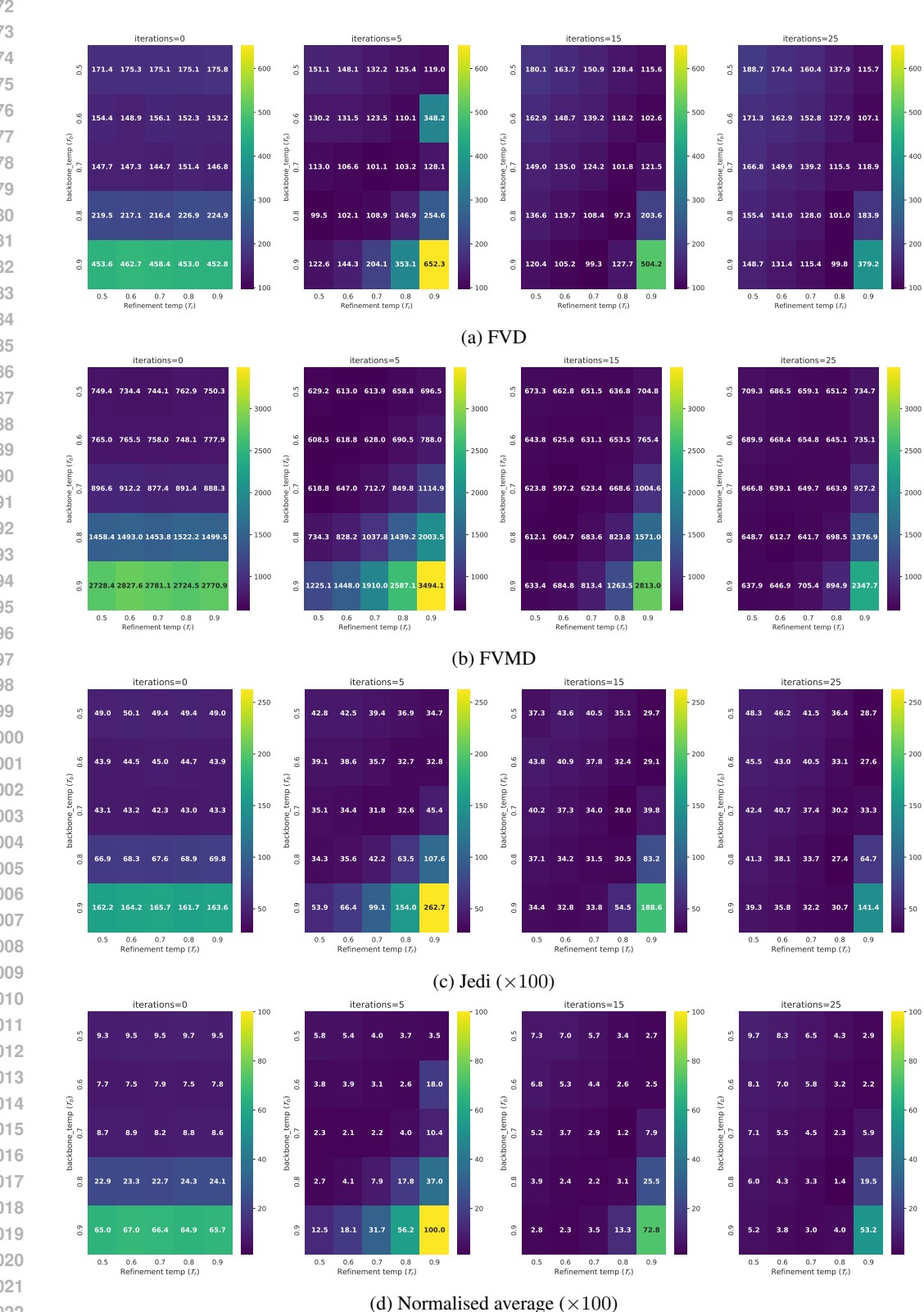

Figure 10: Detailed quantitative results: heatmaps reporting computed metrics for all temperature and iteration settings.

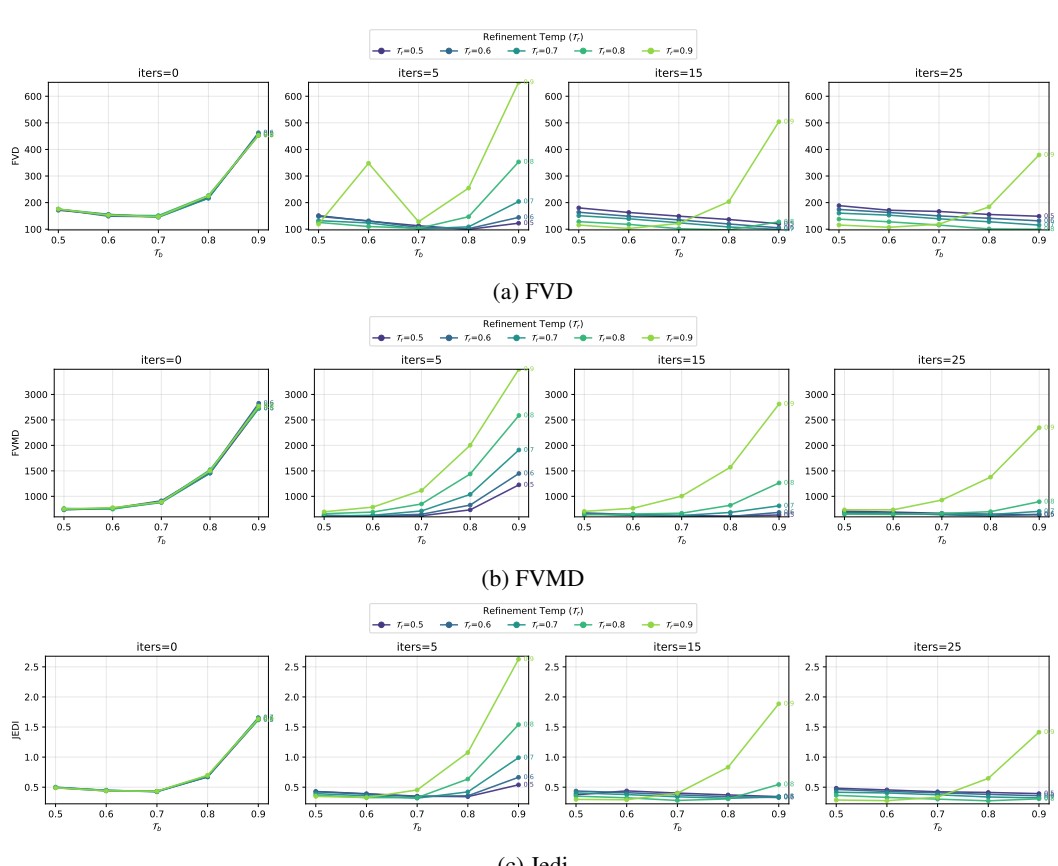

(a) FVD

(b) FVMD

(c) Jedi

Figure 11: Per metric results. We show individual metric results and the impact of temperature and iteration values on overall performance.

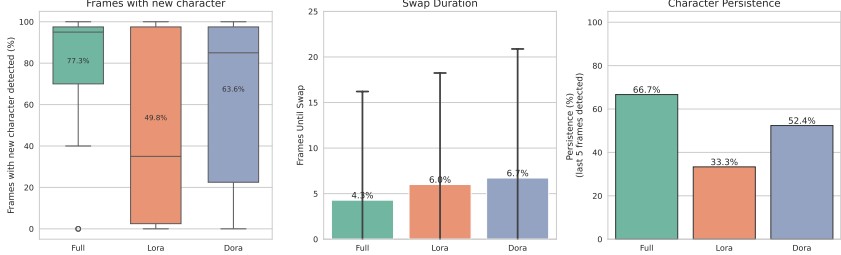

Figure 12: Quantitative Bodyswap results for Fully fine-tuned and PEFT models, using ambiguous prompts. We measure Swap success rate as the number of generated frames with the right character, Swap duration by counting the number of frames until the new character is recognised and character persistence by measuring if the correct character is recognised in the last 5 frames.

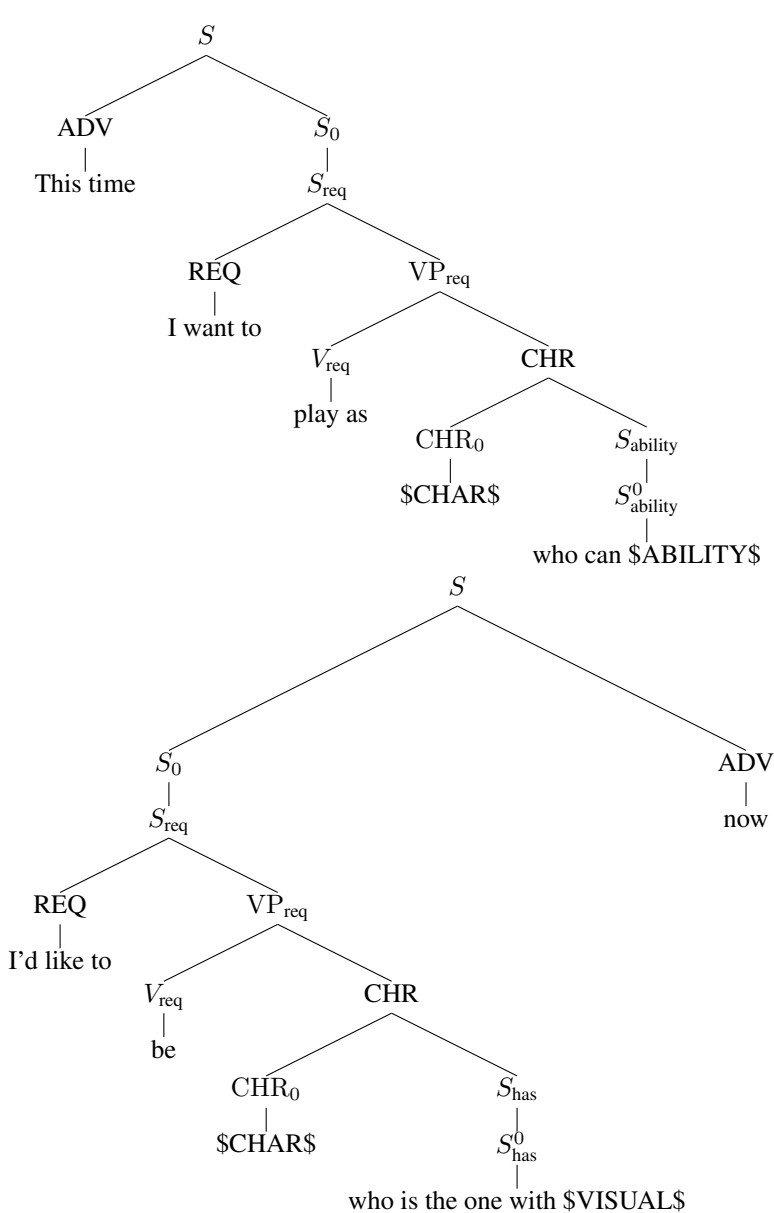

Figure 13: Examples of samples from the context-free grammar

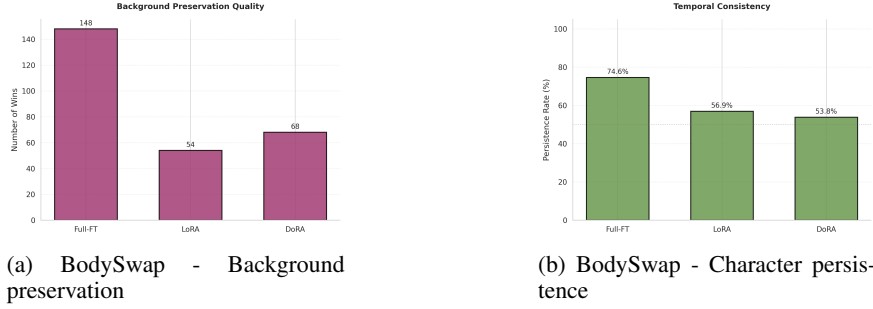

(a) BodySwap - Background preservation

(b) BodySwap - Character persistence

Figure 14: Human Evaluation - Additional BodySwap results

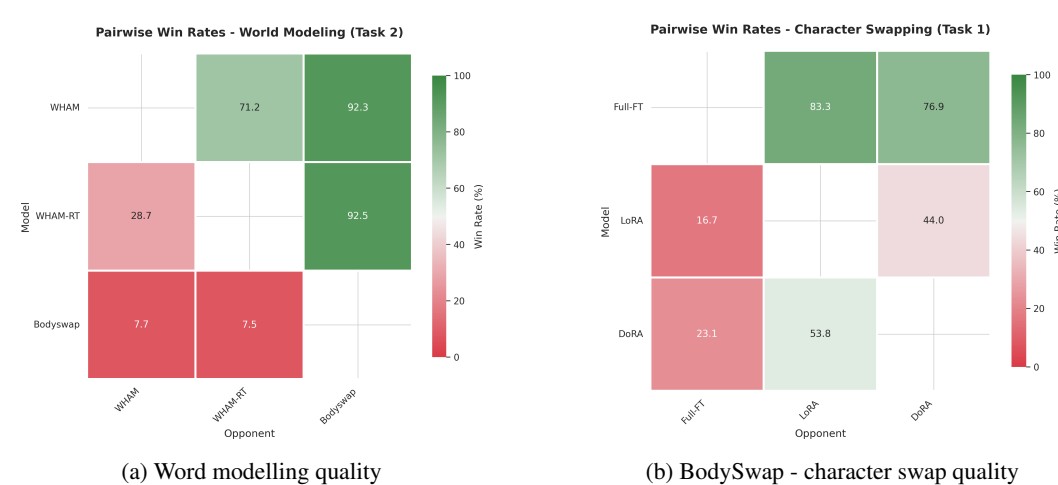

(a) Word modelling quality  (b) BodySwap - character swap quality

Figure 15: Human Evaluation Results - Pairwise win rate

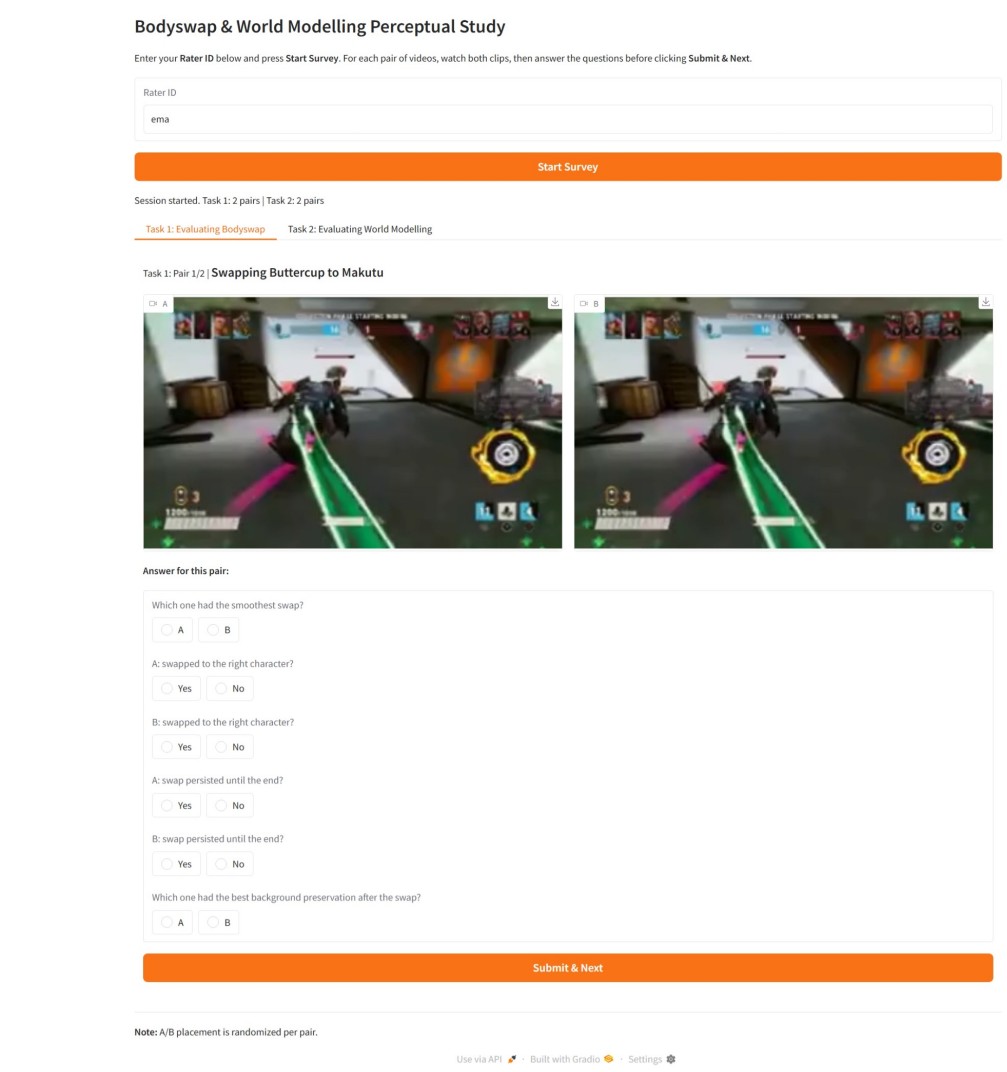

Figure 16: User Interface for our human evaluation study.

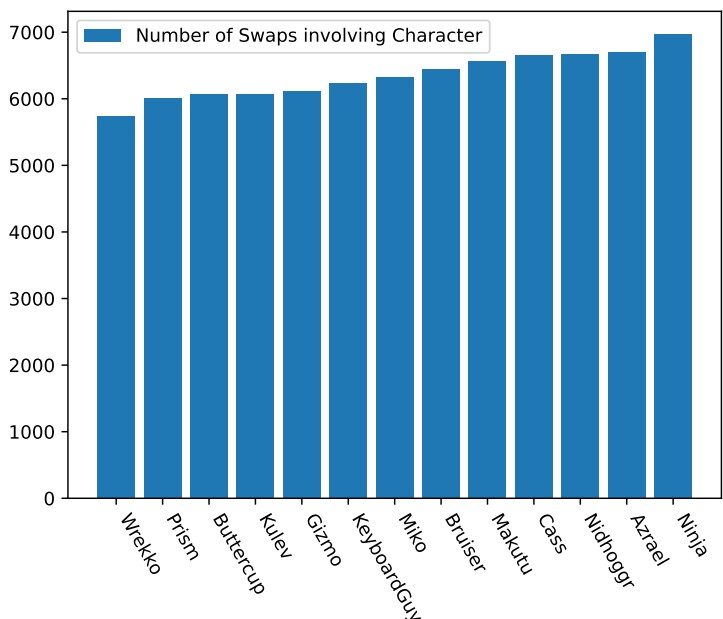

Figure 17: Character distribution in the BodySwap training dataset: number of swaps involving a character.

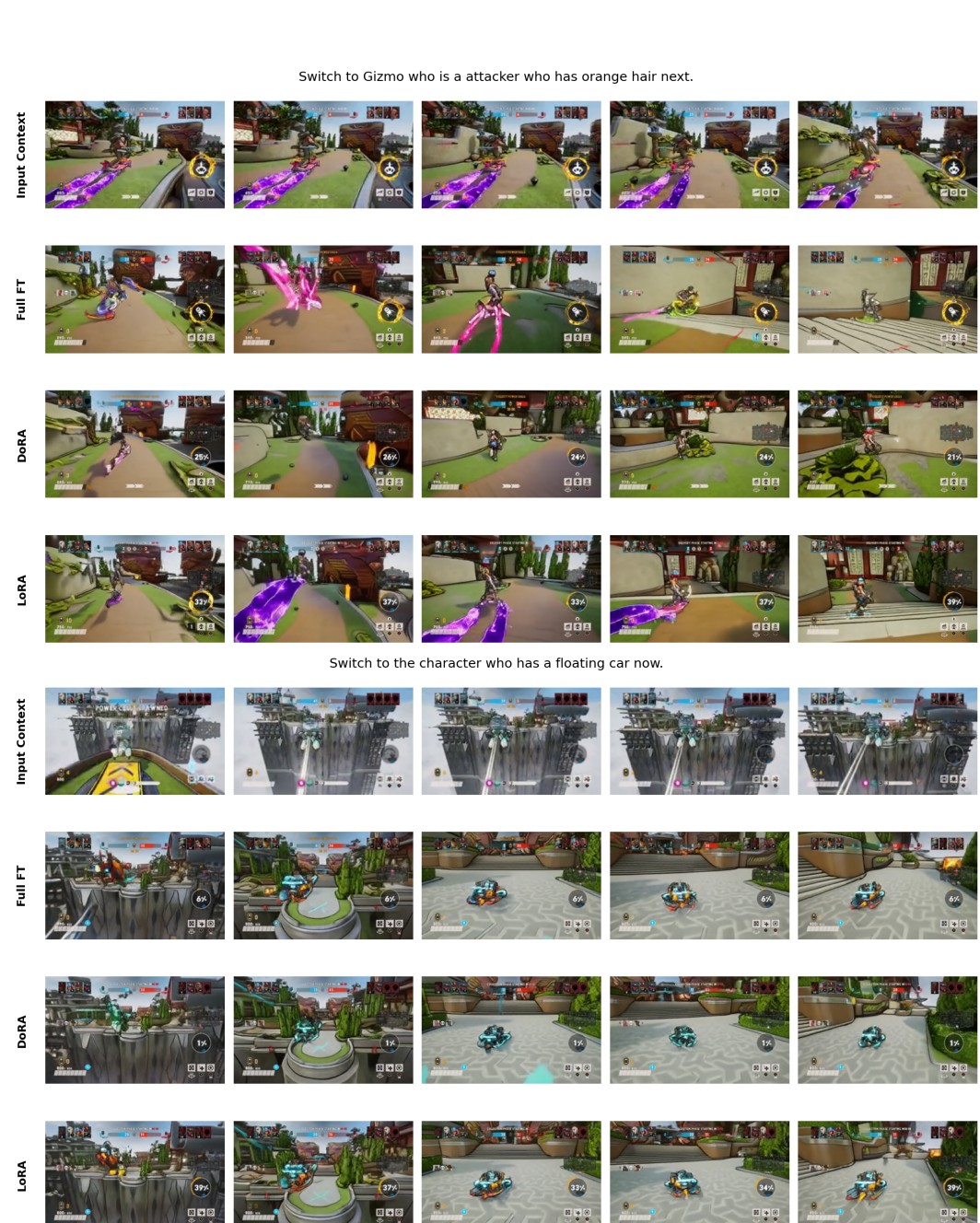

Figure 18: Qualitative BodySwap results - Comparing LoRA, DoRA and Full Finetuning (Full FT) models with the same input context and prompt.

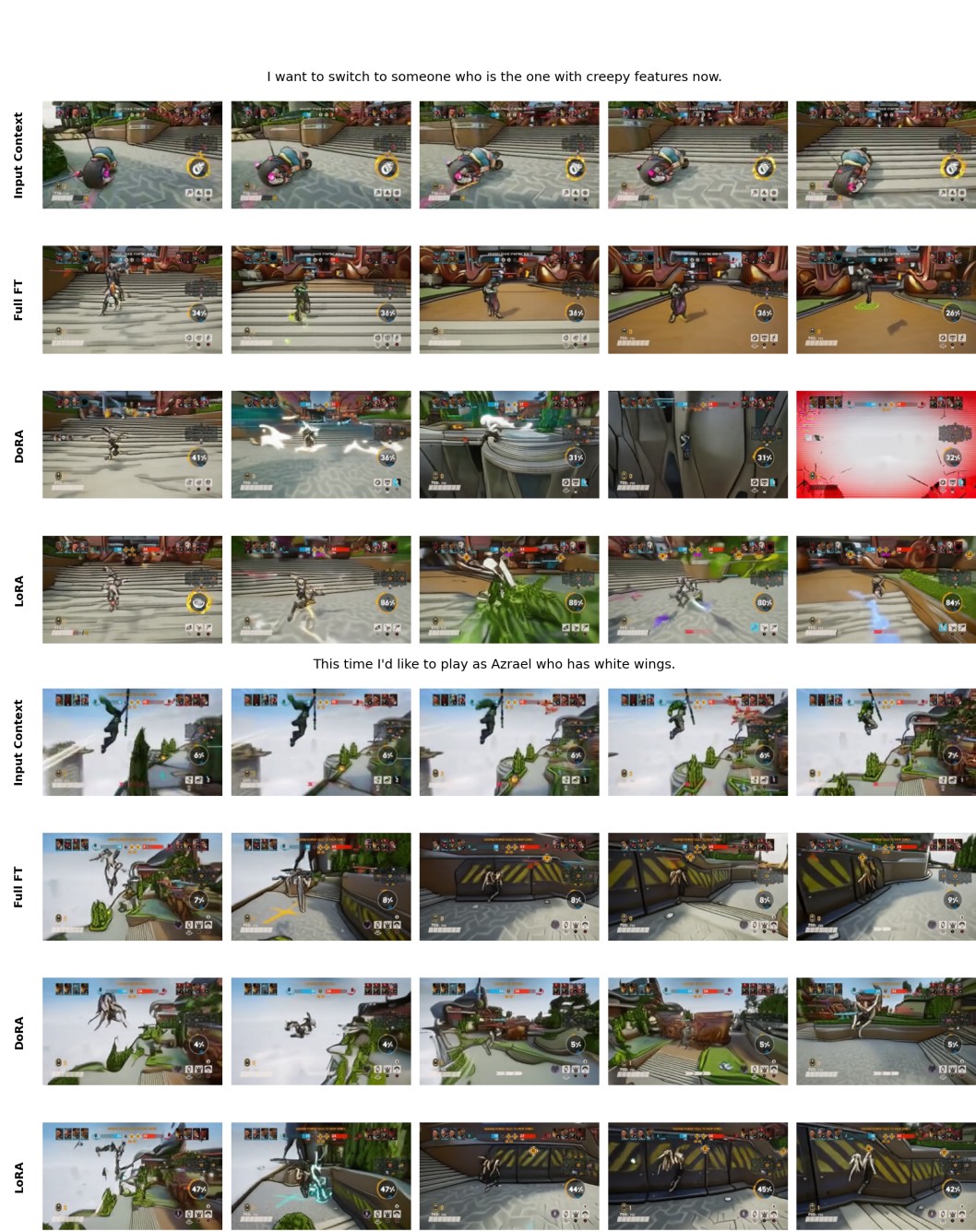

Figure 19: Qualitative BodySwap results - Comparing LoRA, DoRA and Full Finetuning (Full FT) models with the same input context and prompt.

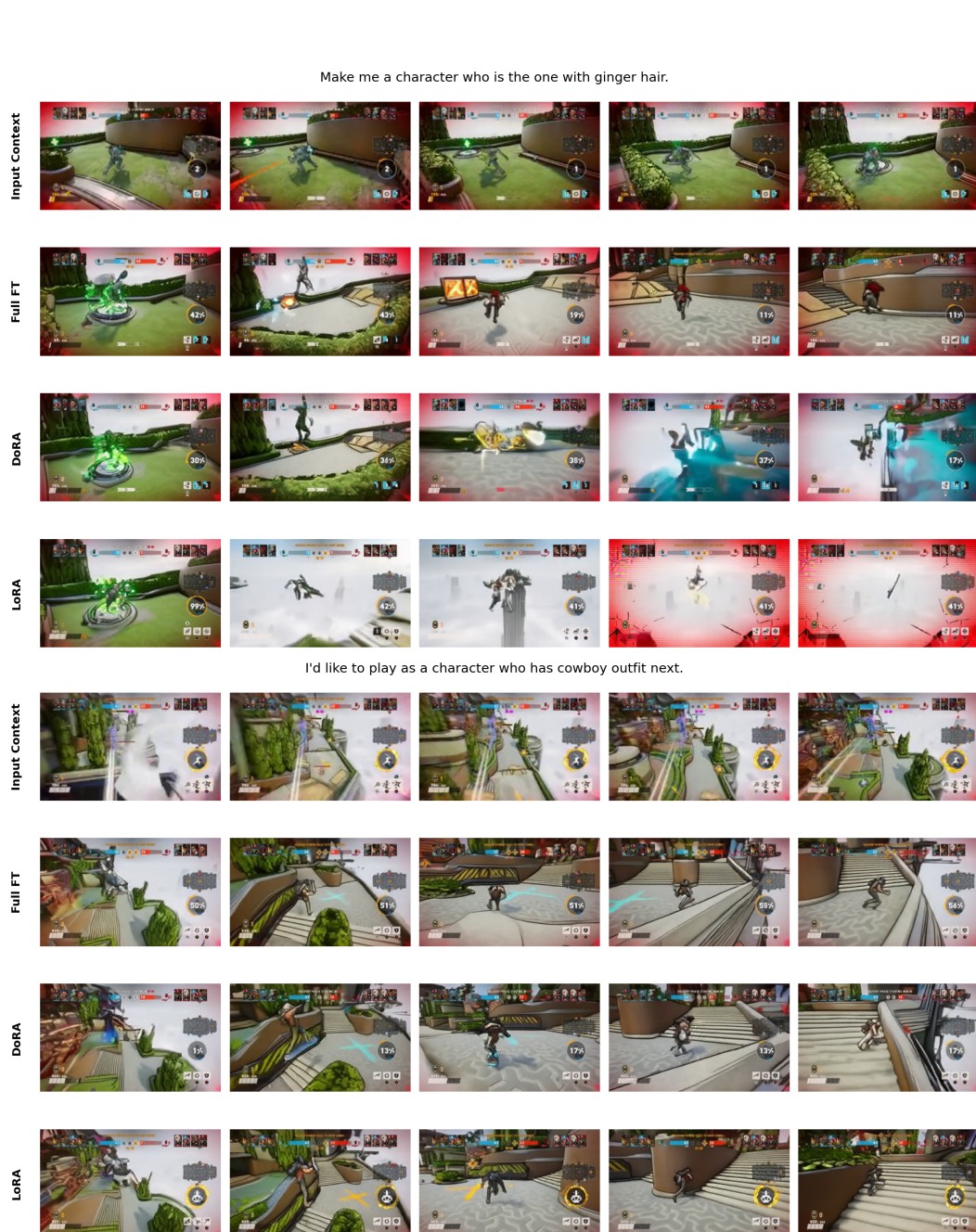

Figure 20: Qualitative BodySwap results - Comparing LoRA, DoRA and Full Finetuning (Full FT) models with the same input context and prompt.

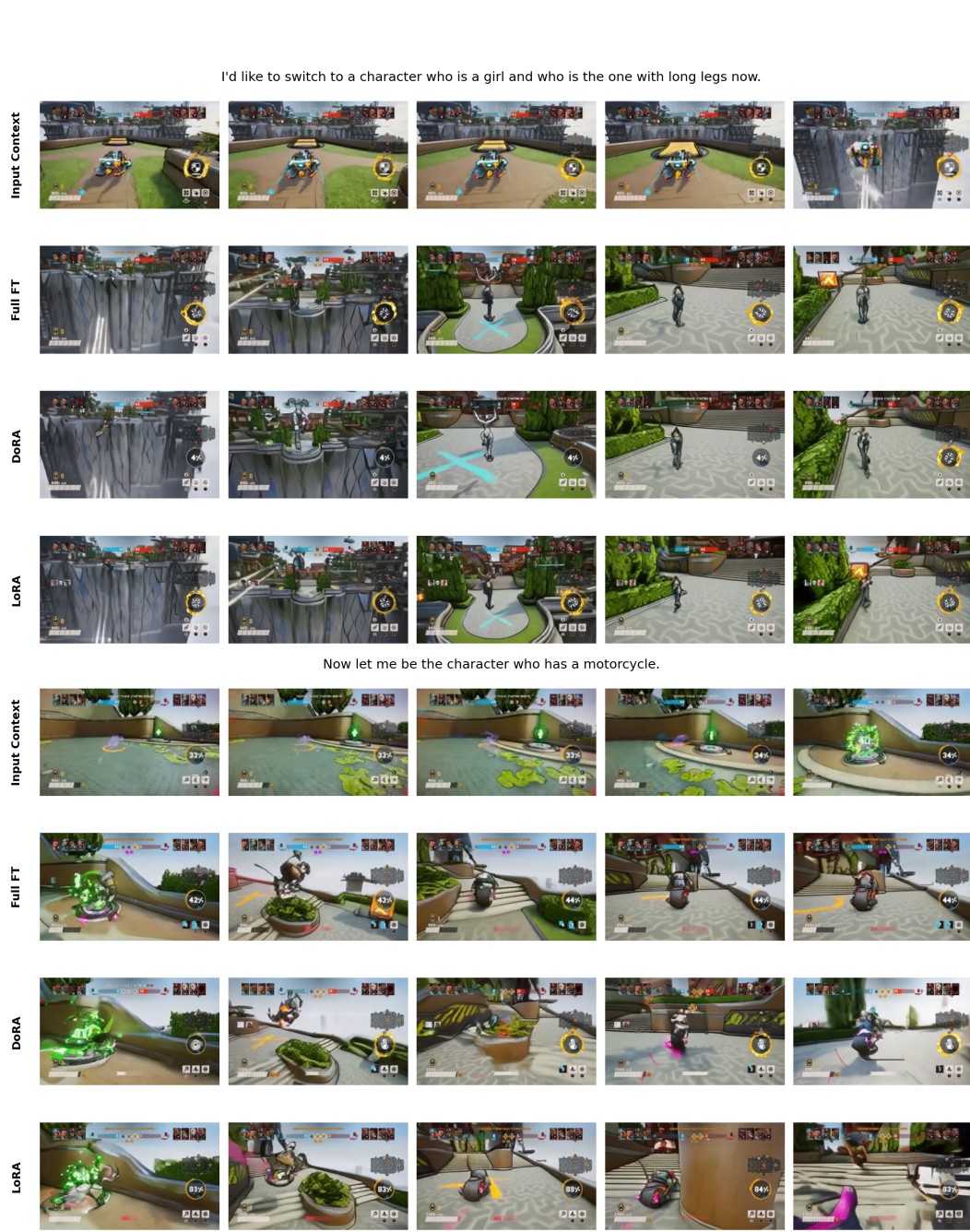

Figure 21: Qualitative BodySwap results - Comparing LoRA, DoRA and Full Finetuning (Full FT) models with the same input context and prompt.

