# OpenReview forum: "Real-Time Text-Conditioned World Models for Interactive Prototyping"
_ICLR.cc/2026/Conference — Submitted to ICLR 2026_

### Official Review · Reviewer_8EMq · 2025-10-30

**Soundness:** 4
**Presentation:** 3
**Contribution:** 3
**Rating:** 6
**Confidence:** 4

**Summary:**

The authors present a new video world-model trained on the game Bleeding Edge which is capable of generating simulated frames in approximately real time (~17 FPS) while maintaining coherence over a moderate time horizon (5 minutes). The authors also introduce a version of the world model which is fine-tuned to swap the in-game character in response to text input. The models are evaluated according to both a variety of automated image quality metrics as well as a human preference study.

**Strengths:**

The technical contribution of this paper seems quite impressive -- an action-conditioned world model which runs efficiently enough to permit real-time play is a real accomplishment and paves the way for future advances in that direction. The architecture appears well motivated and the quantitative results are thorough enough to motivate other researchers to build off of the ideas. The paper is also clearly written and well-explained.

**Weaknesses:**

While I very much appreciate the motivation to begin exploring prototyping and adaptation to novel mechanics with the BodySwap dataset and model, I do also feel that it is the weaker aspect of the paper. The stated motivation is to facilitate prototyping, but the BodySwap model does not simulate any mechanics which have not already been implemented in the existing game environment. That is, while mid-game character swapping does not appear in Bleeding Edge, each of the characters obviously already do. It’s not totally clear that training a world model would be more efficient than implementing the relevant mechanic for the purpose of prototyping. The main upside of a world model in this context (zero- or few-shot simulation of an unimplemented game mechanic) doesn’t seem like it’s fully being realized with the BodySwap example. Even if the results would be less impressive, I think the paper would be strengthened by an exploration of adaptation to truly novel game mechanics (e.g. “swapping” to a character that doesn’t exist).

**Questions:**

- How important is the size of the original WHAM training dataset to the performance of WHAM-RT and BodySwap? This seems like one of the main barriers to wider adoption of the general technique.
- Appendix E only says that “N” human raters were recruited -- what’s the value of N?

---

> ### Author Response · Authors · 2025-11-21
>
> We thank the reviewer for their time and feedback that improves the quality of our work. We address all the reviewer’s concerns below:
>
> -	Bodyswap task too simple
>
> The reviewer raised a valid concern, but as they also noted, we view the BodySwap feature as a proof-of-concept mechanic that is not yet present in the current version of the game. We highlight that our core contribution is not the mechanic itself, which indeed, does not extend beyond the already existing set of game characters, but the idea that new behaviours can be learned cheaply and in a highly controllable manner with intentional and careful data curation. This contrasts with the current standards focusing on training at scale with the highest possible diversity, limiting control over how behaviours are generated. We emphasise our focus on controllability, further enabled by our design allowing to control the trigger of the new mechanic.  We regard extending the world modelling ability to unseen content as future work as it would require training a tokeniser on a more diverse dataset and require more complex data curation tools such as image editing, which is beyond the scope of our study.
>
> -	Size of training datasets
>
> For broader adoption, we would like to emphasise that both the WHAM-RT model and the BodySwap dataset will be open-sourced to support reproducibility and further development.
> To capture a wide range of perspectives within a world, it is important for the initial training dataset to include a wide variety of data points. The WHAM-RT model was trained on a dataset of approximately 310 million frames. In comparison, the BodySwap dataset used for finetuning is much smaller, containing only 41,260 training samples (Appendix F), each lasting 5 seconds—this totals about 2 million frames at a rate of 10 frames per second. In subsequent WHAM-RT experiments, we have explored training it on a much smaller dataset of 1 million frames with promising initial world modelling results which indicate the dataset requirements could be adjusted to accommodate smaller amounts of data.
>
> -	Value of N
>
> That is a valid point, thank you for pointing this out. The value of human evaluators N is 10 and we have edited Appendix E to reflect that in our latest revision.

---

### Official Review · Reviewer_mqCL · 2025-10-30

**Soundness:** 2
**Presentation:** 1
**Contribution:** 1
**Rating:** 2
**Confidence:** 4

**Summary:**

The paper made some acceleration improvements to WHAM and then collected a BodySwap dataset to fine-tune the model with text control.

**Strengths:**

The proposed method seems reasonable and indeed effective in terms of acceleration.

**Weaknesses:**

- (1) The title of the paper is overly ambitious, as the authors aim to achieve "Interactive Prototyping," but the techniques discussed in the paper are not sufficient to support this goal. Key issues such as the model's generative capabilities, understanding, and memory were not addressed, all of which are crucial for achieving interactive prototyping.
- (2) The acceleration and text control fine-tuning methods proposed in the paper lack significant novelty; they are merely common techniques applied to next-token-prediction models.
- (3) The paper's qualitative results are poorly presented. It is difficult to assess the quality of the generated gameplay videos directly from the figures shown by the authors. The specific relationships between interactive controls and corresponding video frames are hard to identify, and it's unclear whether the text control was successful. Additionally, the number of qualitative results shown is very limited. This greatly weakens the contribution of the paper and raises doubts about the effectiveness of the proposed method.

**Questions:**

none

---

> ### Author Response · Authors · 2025-11-21
>
> We thank the reviewer for their time and feedback that improves the quality of our work. We address all the reviewer’s concerns below:
>
> -	Title
>
> We agree with the reviewer that our work has not solved interactive prototyping with world models, but is taking steps towards achieving this long term goal. Our title, and remainder of the paper, are written so as to refer to interactive prototyping as an objective rather than a solved problem, we do not claim to have solved the task. To improve clarity, we are happy to alter the title as “Real-time text-conditioned world models towards interactive prototyping”.
>
> -	Novelty
>
> We respectfully disagree with the reviewer that our work does not comprise significant novelty. We firstly highlight that while WHAM uses next-token prediction, our WHAM-RT uses a different inference paradigm. To the best of our knowledge, real-time streaming video models (and in particular interactive video models) are typically diffusion models, achieving real-time inference using techniques that reduce the number of inference steps such as distillation. In contrast, we introduce techniques to accelerate a discrete prediction model, without reducing the number of inference steps. This is achieved by compressing the number of context tokens using registers, adapting our backbone transformer to predict a first draft of all tokens simultaneously and introducing a lightweight refinement head where iterations are carried out at very low cost. If the reviewer feels the work lacks novelty, we would appreciate references to prior work with similar contributions so we can better contextualise our contributions.
>
> -	Qualitative results
>
> We agree with the reviewer that it is difficult to interpret generated videos as figures in a manuscript. This is why we included numerous videos showing our world model’s ability, as well as numerous BodySwap examples. We have now added failure modes as well. We appreciate that verifying the accuracy of our BodySwap model can require character knowledge and have included new examples in 4 figures in Appendix G with easily identifiable examples based on character attributes. To further support understanding the accuracy of the model, we have a thorough quantitative and human evaluation highlighting the models’ success rates.

---

### Official Review · Reviewer_XsJT · 2025-10-31

**Soundness:** 2
**Presentation:** 2
**Contribution:** 2
**Rating:** 2
**Confidence:** 5

**Summary:**

This paper presents WHAM-RT, a real-time extension of the WHAM world model for interactive gameplay prototyping, achieving up to 17 FPS (≈7000% speed-up) through a discrete diffusion-style refinement that replaces next-token autoregression while preserving visual fidelity. To enhance controllability, the authors introduce the BodySwap dataset, simulating character-swapping in the Bleeding Edge game and enabling text-based prompts to trigger new gameplay behaviours. Experiments show that novel mechanics can be efficiently learned from small curated datasets via lightweight finetuning methods (Full, LoRA, DoRA), with quantitative (FVD, FVMD, JEDI) and human evaluations confirming a strong balance between generation speed and quality.

**Strengths:**

This paper introduces WHAM-RT, a real-time variant of the WHAM world model, aimed at enabling interactive gameplay prototyping through efficient video generation and text-conditioned control.

The authors replace the original next-token autoregressive prediction with a discrete diffusion-style refinement process, achieving up to 17 FPS (≈ 7000 % speed-up) over WHAM with minimal loss in visual fidelity.

To further expand controllability, the authors curate the BodySwap dataset, simulating character-swapping behaviour in the Bleeding Edge game, and integrate text-based prompts to trigger new gameplay mechanics.

**Weaknesses:**

The architectural modification largely adapts known ideas from MaskGit / MAGVIT; the conceptual leap from diffusion-based image refinement to world-model token generation is incremental.

All experiments rely on the single Bleeding Edge environment; no evidence is given for cross-environment generalisation. The author can provide more game scenes, more diverse combinations of objects and characters to verify the generalization and robustness of the method.

While numerical metrics are extensive, visual comparisons (e.g., temporal coherence heatmaps or failure cases) are scarce. The visual results presented in the paper are not convincing enough; more and higher-quality visualizations should be provided.

The use of proprietary gameplay footage (Bleeding Edge) raises unresolved issues around licensing and dataset release.The description of the dataset in the article is rather vague. Could there be more fine-grained dataset analysis and statistics, as well as more visualizations of the dataset and prompts? This would make it more convenient to evaluate the model's performance based on the training dataset.

The paper is poorly written, with many sections that are difficult to follow. There are numerous typos throughout—for example, in Figure 2, both “tokenizer” and “modeling” are misspelled.

**Questions:**

I believe the current version of the paper does not meet the quality required for acceptance. I recommend that the authors address the identified weaknesses and revise the paper for submission to a future conference.

---

> ### Author Response · Authors · 2025-11-21
>
> We thank the reviewer for their time and feedback that improves the quality of our work. We address all the reviewer’s concerns below:
>
> -	Novelty:
>
> The reviewer claims that our work is a simple adaptation of pre-existing works such as maskgit to world models, and is incremental. We strongly disagree with this claim. While the basic inference paradigm is indeed borrowed from maskgit, the core of our contribution is adapting these techniques to achieve real-time inference with large models and next-frame prediction constraints. This required several non-trivial architectural and methodological changes. As shown in our experiments, a direct application of discrete diffusion is far from being sufficient to achieve the desired inference speed. Some of our key contributions involve: 1) moving the iterative generation procedure to a lightweight refinement head, allowing to carry out multiple masked refinements at substantially reduced costs; 2) adapting the backbone transformer to allow predicting all tokens simultaneously at high quality; 3) compressing the video context using register tokens, further enabling iterative refinements. The large context required for video generation would otherwise substantially slow down the lightweight refinement, preventing from achieving real time performance. All these components were carefully crafted so as to maintain the generation quality while speeding up as much as possible. In addition to this, we were constrained by our alignment with WHAM’s latent space, preventing us from achieving speed-ups by improving the VAE’s compression power.
>
> Lastly, our BodySwap contribution puts the emphasis on introducing targeted new mechanisms, that can be controlled via the introduction of a new modality. We aim to demonstrate  how data can be used as a creative material to test new ideas, showing how this procedure can enable much more control and precision than training large models on internet scale datasets with as much diversity as possible. We consider this way of approaching controllable generative environments to differ from the current standards, and to be a valuable point of view to be shared with the community.
>
> -	Bleeding edge -  single environment
>
> As discussed in our introduction, our work is focused on single environment models, and aims to explore how to generate content beyond what a model was originally trained on through the BodySwap experiment. Our goal is not to build a model that can generate content across multiple environments. Our key point of comparison is WHAM, which is solely trained on the Bleeding Edge model; going beyond this environment would not provide us a valid reference and corresponding results would not be grounded. We highlight that the Bleeding Edge dataset comprises complex actions and interactions with different types of objects and non playable characters. As such, this is a very challenging environment to learn and reproduce faithfully, and constitutes a fantastic test bed for the development of world models.
>
> -	Bleeding edge – proprietary dataset
>
> We strongly agree with the reviewer that the use of proprietary, closed source datasets, can be harmful to academic research. We have worked with WHAM and the Bleeding Edge dataset as our starting point, precisely because of our ability to open-source our work. The Bleeding Edge dataset has been described in detail in [Kanervisto et al., 2025] and the WHAM model is fully open source. We have taken steps towards ensuring the release of the BodySwap data and WHAM-RT model as well to encourage the community to explore new ideas within this environment. We have detailed our prompt construction procedure in our manuscript, and provided visual examples in Figure 2.  We have now expanded our description of the dataset in Appendix F to provide additional details and prompt examples. Please let us know if those details are sufficient or if more information is needed.

---

> > ### Author Response · Authors · 2025-11-21
> >
> > -	Not enough visual results, they are unconvincing and low quality
> >
> > We appreciate the reviewer’s request for more visual results. As it is often unconvincing to show video results in image form in a PDF, we have made efforts to include several videos in our supplementary materials, notably two videos comprising 5 minutes of gameplay, and 17 examples of BodySwap. We have now added 10 more videos, including 5 failure cases as requested by the reviewer. We find these videos to be much more informative than images of frames, but for added clarity, we have included 4 new figures showing comparative BodySwap results across the three models we have trained in Appendix G. These results include failure cases as well. Prompts that refer to  characters visual attributes have been selected so as to facilitate assessment of the success of a swap.
> > Our visual results are supported by robust and thorough quantitative results, as acknowledged by the reviewer. We note that our image resolution is constrained by our alignment with WHAM’s latent space, and that our focus is on being on par with WHAM at substantial acceleration. We would be grateful if the reviewer could specify what additional information and visuals would be required to make our results more convincing to them.
> >
> > -	Paper poorly written and full of typos
> >
> > We thank the reviewer for highlighting inconsistencies in British and American English spelling in Figure 1, which has now been corrected. We disagree however that British English spelling constitutes a typo, and have carefully proof-read the paper to avoid spelling errors. We would be grateful to the reviewer if they could point out actual errors that we have missed.
> > We have aimed to describe our work as clearly as possible, and the quality and clarity of our writing has been acknowledged by Reviewer 8EMq. We would be very happy to clarify and rewrite any sections if the reviewer can share which parts of the paper they found unclear or confusing.

---

### Author Response · Authors · 2025-11-21

We thank the reviewers for their time and feedback and have made efforts to address all raised concerns in individual responses to reviewers.

We have made the following modifications to the manuscript according to comments:

-	Uniformise Figure 1 to British English.
-	Update the title to reflect that interactive prototyping is a long-term objective more clearly.
-	Added 4 figures in Appendix G providing BodySwap visual results comparing all three trained models. Prompts were chosen based on characters visual attributes, to enable the reader to assess the success of the swap.
-	Provided more details about our BodySwap dataset in Appendix F.


We re-emphasise the key contributions of our work towards enabling prototyping with environment specific world models:

-	Achieving real time streaming inference with an autoregressive model. This is achieved by introducing a set of techniques designed to accelerate the discrete  diffusion generation paradigm, such as parallel draft token decoding with learnable input tokens, context compression with register tokens and lightweight iterative refinement with a separate prediction head.
-	We demonstrate that new behaviours can be learned and controlled with small, targeted data curation and fine-tuning, proposing a highly controllable approach that contrasts with standard large-scale training and prompting strategies.
-	We strongly agree with reviewers on the importance of open-research and are committed to open-sourcing our WHAM-RT model and BodySwap dataset to encourage future research and innovation in this field.

We appreciate the reviewers’ insights and hope that the clarifications presented here demonstrate the novelty and significance of our contributions.

---

### Meta-Review · Area_Chair_Nuue · 2026-01-01

**Summary:**

This paper presents WHAM-RT, a real-time extension of the WHAM world model for interactive gameplay prototyping. The main concerns are the use of proprietary gameplay footage, limited experiments, poor writting. A rebuttal is provided to partially address these concerns. But the current version of the paper does not meet the quality required for acceptance. The authors should address the identified weaknesses and revise the paper for submission to a future conference.

**Reviewer Concerns:**

Concerns of Reviewer XsJT is not resolved.
Concerns of Reviewer mqCL is not resolved.
Concerns of Reviewer 8EMq is partially resolved.

**Reviewer Scores:**

Reviewer XsJT would not change their score.
Reviewer mqCL would not change their score.
Reviewer 8EMq would not change their score.

---

### Decision · Program_Chairs · 2026-01-26

Reject